# Cost-effectiveness of vector control for supplementing mass drug administration for eliminating lymphatic filariasis in India

Donald S. Shepard[1]*, Aung K. Lwin[1], Sunish I. Pulikkottil[2,3], Mariapillai Kalimuthu[2], Natarajan Arunachalam[2], Brij K. Tyagi[2,4], Graham B. White[5]

**1** Heller School for Social Policy and Management, MS035, Brandeis University, Waltham, Massachusetts, United States of America, **2** Indian Council of Medical Research (ICMR) Centre for Research in Medical Entomology, Field Station, Chinna Chokkikulam, Madurai, Tamil Nadu, India, **3** Regional Medical Research Centre (ICMR), Port Blair, Andaman & Nicobar Islands, India, **4** Department of Zoology & Environment Science, Punjabi University, Patiala, Punjab, India, **5** Department of Entomology and Nematology, University of Florida, Gainesville, Florida, United States of America

* shepard@brandeis.edu

**Data Availability Statement:** All outcome data by village and year are available in the Supporting information accompanying this manuscript to be

## Abstract

### Background/Methodology

Despite progress using mass drug administration (MDA), lymphatic filariasis (LF) remains a major public health issue in India. Vector control could potentially augment MDA towards LF elimination. We conducted a cost-effectiveness analysis of MDA alone and MDA together with vector control single (VCS) modality or vector control integrated (VCI) modalities. Data came from historical controls and a three-arm cluster randomized trial of 36 villages at risk of LF transmission in Tamil Nadu, India. The arms were: MDA alone (the standard of care); MDA plus VCS (expanded polystyrene beads covering the water surface in wells and cesspits to suppress the filariasis vector mosquito *Culex quinquefasciatus*); and MDA plus VCI (VCS plus insecticidal pyrethroid-impregnated curtains [over windows, doors, and eaves). Economic costs in 2010 US$ combined government and community inputs from household to state levels. Outcomes were controlled microfilaria prevalence (MfP) and antigen prevalence (AgP) to conventional elimination targets (MfP<1% or AgP<2%) from 2010 to 2013, and modeled disability adjusted life years (DALYs) averted.

### Principal findings

The estimated annual economic cost per resident was US$0.53 for MDA alone, US$1.02 for VCS, and US$1.83 for VCI. With MDA offered in all arms, all arms reduced LF prevalence substantially from 2010 to 2013. MDA proved highly cost effective at $112 per DALY averted, a very small (8%) share of India's then per capita Gross Domestic Product. Progress towards elimination was comparable across all three study arms.

made available online as Supporting information S3 Table and S4 Table.

**Funding:** DSS, AKL, SIP, and MK received salary support from the Bill & Melinda Gates Foundation (gatesfoundation.org) through grant OPP 43922, 'Resolving the Critical Challenges Now Facing the Global Programme to Eliminate Lymphatic Filariasis' administered by the Task Force for Global Health, Decatur, Georgia, U.S.A. SIP, NA and BKT received salary support from the Government of India through the Indian Council on Medical Research Centre for Research on Medical Entomology (ICMR-CRME, vcrc.icmr.org.in). The funders had no role in study design, data collection and analysis, decision to publish, or preparation of the manuscript.

**Competing interests:** I have read the journal's policy and the authors of this manuscript have the following competing interests: DSS received institutional grant funding from Takeda Vaccines, Inc., Sanofi, and Abbott, Inc. not related to this study. Other authors declare no completing interests.

## Conclusions

The well-functioning MDA program proved effective and very cost-effective for eliminating LF, leaving little scope for further improvement. Supplementary vector control demonstrated no statistically significant additional benefit on MfP or AgP in this trial.

## Author summary

Lymphatic filariasis (LF) is one of the twenty neglected tropical diseases (NTDs) that affect more than one billion people worldwide. Aligned with global initiatives to eliminate LF, India initiated Mass Drug Administration (MDA) trials in limited population groups in the 1980s. Since 2004, the Indian government has implemented repeated MDA campaigns to further this goal. Despite these efforts, LF has not yet been eliminated. Although vector control (VC) is proposed to augment regular MDA to help eliminate LF, little is known about the increased impact or costs. Our study compares the costs of MDA alone to the combination of MDA with alternative VC interventions. For each intervention, we calculated both program operating costs and costs to communities. We found MDA to be very effective (better than initially expected) and cost-effective for eliminating LF. However, against the low endemicity at the study site in the study's final year from MDA alone, the study had limited ability to detect any possible further reductions in MfP or AgP. We found no significant incremental improvements from VC in our study's setting compared to MDA.

## Introduction

Lymphatic filariasis (LF) is one of the twenty neglected tropical diseases (NTDs) that affect more than one billion people worldwide. Aligned with global initiatives to eliminate LF, India initiated Mass Drug Administration (MDA) trials in limited population groups in the 1980s [1]. Since 2004, the Indian government has implemented repeated MDA campaigns to further this goal [2]. Nevertheless, LF remains a public health problem in India [3–7]. This chronic disease is characterized by acute episodes of fever with painful lymph nodes (adenolymphangitis) and progressive swelling of lower and/or upper limbs (lymphoedema) and testes (hydrocele), resulting in loss of labor and worse quality of life for those infected [8]. After decades of work, the latest data from the Global Programme to Eliminate Lymphatic Filariasis show substantial reductions in LF prevalence in most endemic countries, including India [4], and LF elimination from 58 countries as a public health problem. Improved countries include Bangladesh, Maldives, Sri Lanka and Thailand in the South-East Asian Region of the World Health Organization where India has the largest population at risk of LF: 404 million persons [9]. Nevertheless, LF still persists as a global problem [10]. To prevent this debilitating disease in India, the country launched the National Filaria Control Programme in 1955 and targeted the elimination of LF in 2004 [11]. Elimination of LF was based on mass drug administration (MDA), where local workers administer an annual dose of preventive medicines to all residents of the target area (see Annex for acronyms).

 The Indian State of Tamil Nadu, with its strong public health infrastructure, began MDA in 1996 and gradually extended the program to cover all LF endemic districts [6]. The MDA program initially administered a single dose of diethylcarbamazine (DEC) to each eligible person

and subsequently administered DEC plus albendazole annually to the whole population of districts (implementation units) where LF was endemic [12,13]. In 2015, MDA covered 255 districts across 21 states and territories in India, where 630 million people were at risk of LF transmission [3]. However, even after five rounds of MDA with both drugs, pockets of LF infection persisted [14], attributed to inadequate MDA coverage, likely due to non-compliance with timely ingestion of the free antifilarial drugs [15,16]. LF is mostly caused by the nematode worm *Wuchereria bancrofti*, transmitted in India by vector *Culex quinquefasciatus* mosquitoes [17,18] which develop in polluted water, such as water drained into cesspits and sewers [19].

Implementing vector control (VC) programs to interrupt filariasis transmission was considered a potentially beneficial supplement to MDA programs [20,21], perhaps essential towards the elimination of LF [22,23]. Epidemiological evidence of the benefit from supplementary VC [24,25] reported that two annual rounds of MDA augmented by VC with expanded polystyrene beads (EPB) and larvivorous fish to control *Culex* aquatic stages reduced both the annual transmission potential and the transmission intensity index of LF to zero in Tamil Nadu villages [26]. A previous cost-effectiveness study determined that adding VC with two annual rounds of MDA decreased the prevalence of microfilaremia by 1% with an added cost of US$0.85 per person annually [27].

Despite decades of effort, in 2009 LF continued to affect the most marginalized communities. It was not clear whether the existing strategy, MDA, would be sufficient to eliminate LF as a public health problem. The Indian Council of Medical Research Centre for Research in Medical Entomology (ICMR-CRME) then commenced a multi-year prospective randomized trial in Tamil Nadu to examine the effects of specific VC interventions for augmenting MDA, compared with the regular MDA program alone. To our knowledge this study is the only controlled study of this topic. The subsequent results from Sunish et al. [28,29] showed that all of the interventions were associated with significant reductions in LF. Additionally, both VC and MDA strengthened social cohesion by combining community-based initiatives, multiple levels of Indian government, and global partners.

However, sound public health policies depend not only on effectiveness, but also on costs. This lesson was reinforced as emerging problems, such as controlling the COVID-19 pandemic, increased competition for global public health resources. Cost constraints affect all levels—local communities, national governments, and global institutions. The World Health Organization (WHO) described how a well-informed investment plan should enable that organization to meet the challenges of both emerging and long- standing health problems [30].

To determine the best use of available resources for LF, this cost-effectiveness study assessed the costs of the comparative interventions of Sunish et al. and built on previous research by ICMR-CRME [24,25,31]. Our study measured the costs and effectiveness of MDA, assessed the incremental costs and effectiveness of VC interventions, and computed the incremental cost-effectiveness ratios of MDA alone and of adding VC interventions compared to regular MDA alone. Combining detailed cost data with further analysis of outcomes from the cluster randomized trial, the study also offers information for future LF control.

## Methods

### Setting

The State of Tamil Nadu had a population of 72 million persons in 2011 [32]. Lymphatic filariasis was endemic in at least 20 of its 33 districts [33]. The State Health Department (SHD) funds health services in all districts. In 2008–09, the public system contained 42 health unit districts (HUDs) which supervised 1,421 primary health care (PHC) units in Tamil Nadu, with

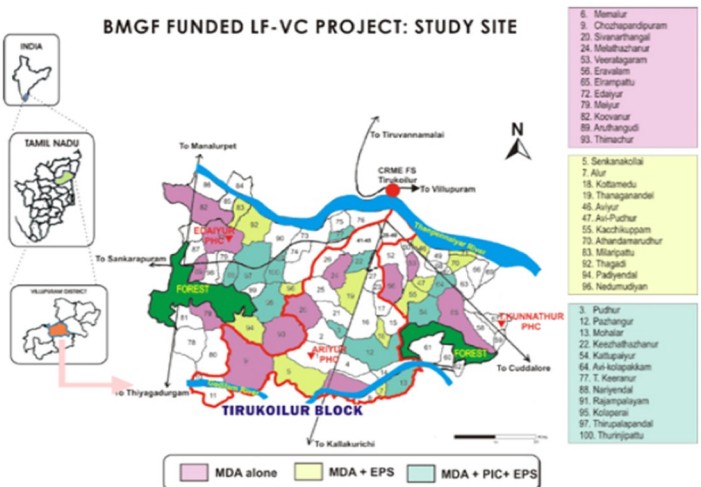

**Fig 1. Map of study villages by treatment arm in lymphatic filariasis vector control (LF-VC) project funded by the Bill & Melinda Gates Foundation (BMGF).** Notes: This map was created by Dr. I. P. Sunish and his staff at CRME-ICMR before the year 2010. It was drawn by hand, without the use of any externally supplied base map. Dr. Sunish has authorized its use here.

each PHC serving at least 30,000 residents [34]. This cluster randomized trial collected data from 2010 to 2013 in areas now forming Kallakurichi and Viluppurum Districts [35]. In 2011, the populations of these areas were 1.35 and 3.46 million residents, respectively [32]. The ICMR-CRME, with headquarters in Madurai, Tamil Nadu, opened a field station in the Tirukoilur block of the Kallakurichi HUD [24]. The ICMR-CRME entomological team selected 36 villages as study sites based on their high risk of LF (Fig 1). Their populations averaged 1,914 residents with a standard deviation of 897. They were under three PHC units in the Tirukoilur block: Ariyur, Edaiyur and T.Kunnathur. Altogether, these three PHCs provided health care for about 150,000 people.

The ICMR-CRME randomly assigned 12 villages to each of the three study conditions. The allocation was balanced exactly by LF MfP and ecological circumstances for mosquitoes. Each condition had 3 villages with MfP<1%, 3 villages with MfP 1–2%, and 6 villages with MfP>2%; they agreed closely in distribution by condition within each PHC block. The World Health Organization [36] defined the eligibility of communities for MDA treatment as the MfP prevalence >1% or the antigen prevalence (AgP) >2%. All villages included in this study were eligible for treatment with the WHO's then recommended two-drug regimen of DEC and albendazole [37]. The initial proportion of tested residents with prevalence above the eligibility cutoff averaged 85% with a standard deviation of 9%.

Kallakurichi HUD lies in northern Tamil Nadu, 40 to 80 kilometers inland from Puducherry on the Indian east coast. The general climate in the region is hot and dry, with moderate rainfall from September to December [38]. The majority of villagers are landless laborers who earn their living from agriculture and livestock [31]. For community management and administration, village committees are led by elected village presidents. Contact between government health personnel and villagers relies on the village health nurse, who manages a PHC satellite unit called a sub-center, that serves approximately 5,000 inhabitants in two to three villages [39].

Prior to the study, no systematic VC operations had been routinely implemented in the study area. Villages have several abandoned wells since tap water standpipes had been

installed. Sanitation involves pit latrines. The majority of houses lack effective barriers to insects, allowing endophilic mosquitoes to easily enter the dwellings and bite residents inside. Due to the region's hot and humid weather, villagers mostly prefer not to use bed nets for the prevention of mosquito bites, because they reduce ventilation. Traditionally, water used for showering and cleaning utensils is disposed of in soakaway pits and wet ditches, which readily serve as breeding sites for *Culex* mosquitoes.

## Description of treatment conditions

The MDA program was intended to occur annually, but actual implementation was adjusted for administrative and logistical reasons. The program administered anti-filarial drugs with DEC and anthelminthic albendazole to target at-risk communities. DEC, by immune mechanisms still largely undefined, causes rapid clearance and destruction of the filarial larvae (microfilariae) circulating in the blood as well as the slower, progressive death of the adult filarial worms. Albendazole is a broad-spectrum anthelminthic drug that principally affects the adult worms, thereby inhibiting the production of microfilariae and suppressing transmission of LF.

In villages assigned VC with a single modality (VCS), a floating layer of expanded polystyrene beads (EPB) was applied at rates of 350–400 g/m$^2$ water surface area over the surfaces of cesspits and unused wells. These concentrations were based on ICMR-CRME's long-standing research [31]. The EPB were designed treat the larval and pupal stages of *Culex* mosquitoes, the main sources of *C. quinquefasciatus*, by deterring mosquito oviposition and asphyxiating the developmental stages by keeping them submerged [40].

While earlier Indian mosquito control efforts utilized the larvicidal bacterium *Bacillus sphaericus*, our study explored the novel application of polystyrene granules. This approach proved both safer and more effective in the village context, where small, contained water sources are common [41]. Polystyrene resin granules (Shri Sakthi Insulations, Chennai, TN, India) were boiled to produce the expanded lightweight ~2mm diameter EPB [42]. To facilitate the placement and retention of EPB in mosquito breeding sites, it was necessary to modify cesspit frames and unused wells to prevent water overflowing, which would flush away the EPBs. Therefore, cesspits were lined with brick walls, and unused wells were cleaned and covered by a custom-made metal mesh screen in a wooden frame.

In villages assigned VC with integrated (VCI) modalities, *EPB* was combined with an additional modality, pyrethroid impregnated curtains (PIC). These treated curtains were installed to mosquito landing sites around houses (doors, eaves, and windows) to kill adult insects by contact with insecticidal pyrethroid [43].

The PIC intervention required measurement of the doors, windows and eaves for tailoring curtains to fit these spaces. Curtain materials were made of white woven ventilated polypropylene (VIRGO Polymer India Ltd., Kanchipuram, TN, India) [44] impregnated with deltamethrin insecticide (2.5% suspension concentrate, Chemet Wets & Flows Pvt. Ltd., Ahmedabad, Gujarat, India) by soaking and drying in the shade to achieve approximately 25mg ai/m$^2$ dosage, before cutting, sewing borders and installation by hanging with nails. Installed curtains had to be re-treated by spraying every six months to sustain insecticidal efficacy (validated by ICMR-CRME bioassays). ICMR-CRME implemented both interventions in their assigned villages from March to June 2010.

## Delivery of interventions

The SHD of Tamil Nadu implements MDA almost annually with tablets of DEC, procured from local manufacturers in India [6], plus albendazole donated by the pharmaceutical

**Table 1. Management levels supporting the LF mass drug administration (MDA) program in the study area and corresponding populations covered.**

| Management level | Staff | Population covered |
|---|---|---|
| 1. State Health Department (SHD) of Tamil Nadu | Members of Task Force and Advisory Committees; Supervisory Team | 31,250,000 |
| 2. Health Unit District (HUD) of Kallakurichi | HUD supervisory team: Deputy Director of Health Services, District Staff Assistant, District Malaria Officer, District Health Inspectors*, Medical College Entomologists*, HUD Rapid Response Team | 1,589,500 |
| 3. Block Primary Health Center (PHC) of Ariyur | Block PHC supervisory team: Block Medical Officer*, Assistant Medical Officer at block, Assistant Medical Officer of additional PHC, Block Health Supervisor, Non-medical Supervisor, Block Extension Educator, Community Health Nurse at block PHC, Sector Health Nurse at block PHC, Health Inspector (MDA in-charge) at Block Health Supervisor*, Non-medical Supervisor, Block Extension Educator*, Community Health Nurse, Sector Health Nurse, Pharmacist*, Sanitary Worker, Driver; Block PHC Rapid Response Team | 116,822 |
| 4. Subcenter | Village Health Nurse * | 5,000 |
| 5. Village | Involved villagers | 1,639 |
| 6. 50 households | Village volunteers * | 250 |

* Indicates survey respondents. Note: management levels are ordered from highest to lowest. LF denotes lymphatic filariaisis.

company GlaxoSmithKline [45], The program followed procedures and dosage regimens recommended by the WHO. This regimen entailed a single annual co-administration of DEC (6 mg/kg body weight) and albendazole (400 mg) for individuals aged 2 years and above and repeating the annual dose for at least 5 consecutive years in areas where LF is endemic [46]. In the study area, the SHD organized the local program to eliminate lymphatic filariasis through five management levels: the Tamil Nadu SHD, Kallakurichi HUD, primary health centers (PHC), sub-centers, and village delivery units of filariasis prevention assistants.

Drugs for MDA were transported by the SHD to its HUDs, then distributed to their block level PHCs. Village health nurses of sub-centers carried drugs from the block level PHCs to their villages and supervised the most peripheral workers of the delivery chain, the filariasis prevention assistants, who were volunteers. Recruited from village communities, filariasis prevention assistants directly delivered drugs to the households and observed residents swallowing their tablets. In each annual MDA campaign, one day is scheduled for filariasis prevention assistants to deliver to households, and two days are scheduled for tracing and providing treatment of those missed on the preceding day.

Multiple levels of management were required to coordinate delivery of the MDA tablets to residents in the 36 villages investigated. Upper health management levels were involved in more extensive distribution efforts to villages across the State that were not part of this study's population, although the cost shares of higher-level centers were pro-rated based on their total population coverages. Table 1 shows how a village was integrated into the combined State efforts of all health management levels and associated responsibilities.

The VC intervention delivery system for this ICMR-CRME research study was less hierarchical than that of the National MDA program administered by the state. Major activities of added VC interventions were carried out solely by the ICMR-CRME team, in close collaboration with Ariyur block PHC staff members, and other field-level government staff for some part of the monitoring and evaluation efforts. ICMR-CRME recruited volunteers from villages, trained and motivated them for "Vector Control through Community" [28] to facilitate VC activities by community members. While the two higher administrative levels (SHD and HUD) did not perform VC operations during the study, they would manage these activities should the VC programs be adopted beyond a research setting. To make comparisons of interventions more realistic for future policies, we simulated costs of senior management level

efforts in addition to actual efforts of ICMR-CRME and the contributions from its associated government management centers.

## Collection of cost data

For both MDA and VC programs, personnel costs were calculated using time allocation surveys, salary and wage records, reports, and narrative interviews with paid and volunteer workers. MDA progress reports were used to allocate SHD staff time, while time allocation surveys were completed by managers and some community leaders for all levels below the SHD. The salary of each full-time worker and the opportunity cost of each volunteer were collected from administrative records at the SHD, HUD, three block level PHCs, the ICMR-CRME headquarters and its field station. Narrative interviews clarified implementation procedures and tasks of health workers and community volunteers.

Both MDA and VC involved considerable efforts by community members. Current MDA programs engaged full community mobilization efforts once a year, while VC programs required full community mobilization in the initial year and regular maintenance by villagers in subsequent years. During site visits, Brandeis investigators with ICMR-CRME interpreters interviewed participants at every level from the SHD to households.

Non-personnel items consisted mainly of transportation costs (allocated costs of vehicles and fuel) for all activities, drugs for MDA, polystyrene granules, cooking pots and fire for EPB production, construction materials (bricks, cement, tools) for improving cesspits, insecticide, curtains and tools for cutting and sewing them for PIC. We collected information on non-personnel costs of MDA from administrative records from the PHC, interviews with supervisors, and vector control program reports [47]. To estimate economic costs of donated drugs, we asked informants for local retail prices and validated these against the International Drug Price Indicator Guide, 2009 [48]. For VC interventions, ICMR-CRME documents and officials provided expenditures on non-personnel items. Authors communicated regularly with ICMR-CRME field unit staff (Figs A and B in S1 Text).

Our cost analysis further separated financial and non-financial costs by the type of input. Personnel or materials procured specifically for an intervention represent financial costs, while use of existing or donated resources (including donated drugs) are non-financial costs.

## Analysis of cost data

WHO and its international partners, such as the United States Agency for International Development (USAID), adopted a 3-stage "roll-out package" [49] against NTDs, assuming that integration and scale-up of various programs to full-national scale is feasible [50]. Our analysis followed similar assumptions that various approaches to eliminating a single NTD can be integrated and scaled up by the national ministries.

As noted, VCS used EPB, while VCI combined EPB and PIC. We estimated the cost of each intervention separately and added the components to estimate the final costs of the integrated approach. Our analysis of unit costs of each intervention consists of three major steps: (1) estimating total recurrent costs (personnel and non-personnel) by management level, (2) annualizing capital costs, and (3) calculating final unit costs per beneficiary.

We used Microsoft Excel (Redmond, WA, USA) to create two cost analysis tools: mass drug administration cost analysis tool and VC cost analysis tool, which were used to enter data and perform cost analyses. Our data collection instruments quantified personnel activities and non-personnel inputs and their unit costs. We categorized these costs by each management level, i.e., the SHD, intermediary management levels, and household groups. As higher levels served populations of varying sizes beyond the study's 36 villages (Table 1), we converted the

costs at each administrative level to their corresponding per capita amounts, including the administrative costs of hiring personnel and procuring materials for VC and MDA. To compare yearly costs across the implementation strategies, we amortized capital assets over their useful lives. The analysis tools were refined and validated through discussion with Indian government officials.

In contrast to the State-run annual MDA programs, the VC activities were initiated specifically for this study, managed by ICMR-CRME, and funded with donor support. Community mobilization was most intense in the first year; project managers estimated that only 15% of the staff's initial year efforts were necessary in subsequent years when communities were presumed to be self-sufficient for continuing their VC efforts. While MDA required the procurement and dissemination of drugs once a year, the lifetime of installed VC materials varied. For example, insecticide spray on curtains needed to be reapplied every six-months. EPB required yearly production and reapplication, but PIC were expected to last for three years.

Our cost adopted the societal perspective, which encompasses a broader view of the economic impact beyond just the program itself. However, to provide further information to policymakers on program costs, we have separated them between those borne by government (including donors in partnership with government) versus the community, and distinguished financial and non-financial costs.

Our analysis computed per capita costs based on the number of residents benefiting from each service. For MDA interventions, the combination of drugs served one person. For PICs, the curtain benefited one household (average of five persons) while one cesspit modification served 27 households (135 persons). Most cost data on personnel activities and non-personnel inputs were initially collected in Indian rupees. The project was planned, and many costs were incurred in 2010. We therefore converted Indian rupees to US dollars based on the market exchange rate of 45.8 rupees per US dollar in early 2010 [51] and expressed all costs in 2010 US$.

For consistency with recommended guidelines for economic evaluations, we applied a 3% discount rate to the costs incurred in subsequent program years [52]. This discount rate reflects the real time value of money. This discounting makes our findings consistent with most cost-effectiveness studies.

### Analysis of impact

The ICMR-CRME entomological team collected baseline data of parasitological prevalence and entomological status before the intervention (February 2010) and continued to collect yearly data on the post-intervention progress for three years. We collected data on microfilaria prevalence (MfP) tested by microscopic examination of night-time collection of blood smear samples and AgP using the immunochromatography card test (BinaxNOW Filariasis, Alere Inc., Portland, Maine, USA) for MfP and AgP, commonly used indicators to monitor progress towards LF elimination [53–55].

We estimated the number of infections prevented by comparing LF prevalence before and after interventions. To evaluate the impact of the VC strategies, we first evaluated the impact of the reference strategy, MDA. As MDA is the standard treatment in India, there was no control area without MDA. Instead, we selected as the comparator the historical situation in 2010, as implementation of and compliance with MDA had traditionally been low [56]. We calculated the cumulative improvement from 2010 to 2013 in the share of villages reaching the benchmarks towards elimination of MfP and AgP as public health problems—MfP<1% and AgP<2%. Dividing the cumulative improvement by 3 (the number of years), we calculated the average annual improvement.

To evaluate the incremental improvement of VC over MDA, we performed logistic regression analyses with these shares of villages controlled based on MfP<1% or AgP<2% as the dependent variable using Stata version 15 (StataCorp, College Station, TX, USA). The independent variables were the final year of intervention (year), the intervention (MDA alone, as the baseline level, VCS, and VCI), and interaction terms. These interaction terms on MfP and AgP (intervention level x year) are the main variables of interest, indicating the differential effectiveness of each combination of interventions after controlling for baseline conditions and general trends. Finally, to contextualize our findings we examined India's and the state's prevalence of LF infection from 1990 through 2021, the latest data available.

## Cost-effectiveness analysis

For assessing the cost-effectiveness of VC, we converted the average annual improvements from VC by arm into disability adjusted life years (DALYs) averted per 1,000 population. As the first step, we needed the annual disability adjusted life years (DALYs) burden per person affected by LF. The one study of which we were aware that explicitly gave a value reported an annual burden per case of 0.11 [57]. However, this relatively high value of 40 days of extreme disability per year apparently reflected only symptomatic persons, rather than the broader population of persons infected. To derive an estimate for this broader population, we relied on the Global Burden of Disease (GBD) study. Using the last pre-pandemic year (2019), which we considered representative of the study period, we divided India's rate of years lost to disability (58.96/100,000) due to LF by its prevalence (2.68%) [58], deriving a burden of 0.022 DALYs per person affected. Finally, we divided the incremental cost per 1,000 population by the incremental DALYs averted per 1,000 population to get the incremental cost-effectiveness ratios (ICER) by arm.

## Results

### Economic cost of interventions

We estimated the incremental cost and impact of each VC intervention over the conventional approach, MDA alone. We amortized capital costs and estimated recurrent costs through an "ingredients approach," multiplying the quantity times the unit cost of each input. Key inputs for VC (cesspit modification and curtain material) have useful lives of 3 years, whereas all inputs for MDA have useful lives of only one year. In calculating costs, inputs with longer useful lives have an economic advantage because they do not need to be repeated so frequently. Table 2 summarizes lifespans of the inputs and their target beneficiaries.

At the village level, the only input that MDA requires is personnel, as other inputs come from higher levels. EPB and PIC require both personnel and non-personnel inputs. For both EPB and PIC, non-personnel costs exceeded personnel costs. Table 3 presents the ingredients, their unit costs and the quantities of each component and totals per 1,000 population at the village level. At the village level, the cost of EPB is twice the cost of MDA ($129.5 versus $66.6 per 1,000 population), and PIC is five times the cost of MDA (361.5 versus 66.6).

At a higher (PHC block) level, the intervention costs repeat the pattern from the village level. MDA requires only personnel. Again, for both EPB and PIC, non-personnel costs exceed personnel costs. Also, EPB and PIC again cost more than MDA. At the block level, EPB is triple the cost (339.8 versus 101.7) and PIC is four times the cost of MDA (420.0 versus 101.7). Table 4 presents the cost analysis at the block level.

Table 5 expands the cost analysis per 1,000 population to all levels of the health care system. Its subtotals show that approximately 25% of the economic costs for the MDA program ($110.94 compared to $533.55) were borne by the communities served. From the community perspective, PIC was the costliest intervention ($361.47), EPB was intermediate ($129.46), and

**Table 2. Useful life spans and targets of MDA and VC inputs.**

| Inputs | Useful life (years) | Intervention targets |
|---|---|---|
| Mass drug administration (MDA) | | |
| Sensitization by government. | 1 | All residents |
| Drugs | 1 | All residents |
| Vector control (VC) *Both interventions* | | |
| Peer sensitization of villagers | 5 | All residents |
| *Expanded polystryene beads (EPB)* | | |
| Cesspit modification | 3 | Cesspits (1 pit per 27 households) |
| Application of beads | 1 | Wells (1 well per 55 households) |
| *Pyrethroid impregnated curtains (PIC)* | | |
| Curtain material | 3 | All houses (1 house per 5 persons) |
| Spray | 0.5 | All houses (1 house per 5 persons) |

MDA was least costly ($110.94). For government and community inputs together, the cost of adding VCS (i.e., EPB) to MDA was $490.92 (i.e., $129.46 plus $361.47 with rounding) per 1,000 population. The lower panel of Table 5 shows only a small percentage of MDA costs was financial (2%), compared to most for EPB (92%) and PIC (87%). As VCI combines EPB and PIC, the overall cost to add it to MDA was $1,294.43 (i.e., $490.92 plus $803.51) per 1,000 population.

Fig 2 presents the breakdown of intervention costs by activity and intervention. Among the three interventions, PIC is the costliest. Among activities within each intervention, procurement of the materials and delivery of the service (termed procurement and delivery) was the costliest category. The non-personnel costs of VC were much higher than those of the MDA program. As EPB and PIC are supplements to MDA, each arm with these interventions costs more than MDA alone.

## Progress towards LF elimination in the study area

Descriptive analyses showed a decline in microfilaria prevalence rates (MfP) for village groups within each of the treatment conditions from 2010 through the end of ICMR-CRME's collection of entomological data in February 2013. Under MDA-alone, MfP fell from its baseline levels (± standard error of the mean, using the approximation of a normally distributed variable) by 84 (±25) percentage points. Where MDA was augmented by VC, villages with VCS reduced MfP by 75 (±20) and villages with VCI reduced MfP by 90 (±22) percentage points, respectively. All reductions were statistically significant compared to baseline levels [28]. Similarly, the "disease-free" share of population (i.e., living in villages with MfP below 1%) increased from 17% to 83% (66 percentage-point improvement) in villages with MDA alone, from 8% to 67% (59 percentage-point improvement) in VCS villages, and from 17% to 92% in VCI villages (75 percentage-point improvement).

The percentages of villages with elimination are shown in Fig 3 for MfP and Fig 4 for AgP, with the underlying data in Tables A and B in S1 Text. Fig 5 shows the average AgP by study area and year. Similar to the results for average prevalence, all arms showed substantial improvement over time. However, the ranking of arms varied by year and by indicator, suggesting that no single arm had dramatically superior results.

Using the AgP measure for each intervention, the relative reductions from 2010 to 2013 were similar to those for MfP: 70 percent for MDA-alone, 89 percent for VCS, and 79 percent for VCI.

**Table 3. Annualized costs per 1,000 population at the village level (US$).**

| Input | Unit cost (US$) | Number of units | | | Total costs (US$) | | |
|---|---|---|---|---|---|---|---|
| | | **MDA** | **EPB** | **PIC** | **MDA** | **EPB** | **PIC** |
| Village residents offered MDA (hours) | | | | | | | |
| Village president | 0.56 | 9.76 | | | 5.46 | | |
| Village vice-president | 0.45 | 9.76 | | | 4.37 | | |
| Village ward member | 0.45 | 29.29 | | | 13.10 | | |
| Others (e.g., previous presidents) | 2.24 | 19.52 | | | 43.66 | | |
| Village residents offered EPB & PIC (hours) | | | | | | | |
| *Personnel* | | | | | | | |
| Village president | 0.52 | | 1.17 | 1.58 | | 0.61 | 0.83 |
| Village vice-president | 0.42 | | 0.26 | 0.47 | | 0.11 | 0.20 |
| Village ward member | 0.45 | | 4.08 | 2.58 | | 1.82 | 1.15 |
| Village president's ass't | 0.22 | | 4.18 | 8.00 | | 0.91 | 1.75 |
| Village ward member's ass't | 0.22 | | 11.90 | 16.05 | | 2.60 | 3.50 |
| Household head (procuring products) | 0.22 | | 0.82 | 276.58 | | 0.18 | 60.39 |
| Village laborer | 0.05 | | 2.38 | | | 0.13 | |
| Village meeting participant | 0.14 | | 13.34 | | | 1.87 | |
| Secondary school student | 0.04 | | | 174.68 | | | 7.75 |
| Mason | 0.61 | | 1.77 | | | 1.09 | |
| Mason's helper | 0.31 | | 1.71 | | | 0.52 | |
| Well cleaner | 0.61 | | 3.51 | | | 2.16 | |
| Well cleaner's helper | 0.12 | | 2.11 | | | 0.26 | |
| Village volunteer | 0.22 | | 22.84 | 64.41 | | 4.99 | 14.06 |
| Demonstration participant | 0.22 | | | 13.97 | | | 3.05 |
| PIC impregnator | 0.22 | | | 69.87 | | | 15.25 |
| Neighborhood helper | 0.22 | | | 174.68 | | | 38.14 |
| Other village residents attending demonstration | 0.14 | | | 27.29 | | | 3.82 |
| Tailor | 0.68 | | | 3.49 | | | 2.36 |
| Tailor's helper | 0.23 | | | 3.49 | | | 0.79 |
| Sprayer | 1.53 | | | 9.36 | | | 14.30 |
| *Subtotal personnel* | | | | | 66.6 | 17.2 | 167.4 |
| *Non-personnel* | | | | | | | |
| Expanded beads (kg) | 3.8 | | 28.2 | | | 107.8 | |
| Curtain fabric (m$^2$) | 0.49 | | | 351.6 | | | 173.7 |
| Other for EPB treatment | 0.004 | | 1100 | | | 4.4 | |
| Other for PIC treatment | 0.020 | | | 1025 | | | 20.5 |
| *Subtotal non-personnel* | | | | | 0.00 | 112.2 | 194.1 |
| *Total (all resources)* | | | | | **66.6** | **129.5** | **361.5** |

Notation: MDA denotes mass drug administration; EPB denotes expanded polystyrene beads; PIC denotes pyrethroid impregnated curtains; VC denotes vector control; hr denotes hour; kg denotes kilogram, m$^2$ denotes square meter.

Note: Total costs equal unit cost times the exact quantity. Quantities shown in the table are rounded.

The observed significant reductions in MfP and AgP following the intervention are most likely attributable to the interruption of the LF transmission cycle. The interventions–MDA, VCS and VCI--effectively reduced the parasite reservoir in humans and/or the transmission potential of mosquitoes. This led to a decrease in new infections, ultimately resulting in lower prevalence rates.

**Table 4. Annualized costs (US$) per 1,000 population at the block primary health care level.**

| Input | Unit cost | Number of units | | | Total cost (US$) | | |
|---|---|---|---|---|---|---|---|
| | | MDA | EPB | PIC | MDA | EPB | PIC |
| Block primary health center staff (hours) | | | | | | | |
| Medical officers | $4.8 | 2.3 | 0.4 | 0.4 | 11.3 | 2.1 | 2.1 |
| Public health supervisors | $4.5 | 3.2 | 0.7 | 0.7 | 14.4 | 3.0 | 3.0 |
| Nurses | $4.0 | 2.6 | | | 10.2 | | |
| Health inspectors | $3.5 | 4.7 | 3.0 | 3.0 | 16.5 | 10.7 | 10.7 |
| Assistant field workers | $2.5 | 1.7 | | | 4.1 | | |
| Administration workers | $2.5 | 1.2 | | | 2.9 | | |
| Press people | $2.2 | 0.5 | | | 1.1 | | |
| Personnel others | $5.3 | 7.7 | | | 41.2 | | |
| VC project staff (hours) | | | | | | | |
| Lead persons | $5.5 | | 5.8 | 5.8 | | 31.9 | 31.9 |
| Project officers | $1.6 | | 11.4 | 11.4 | | 17.6 | 17.6 |
| Technicians | $0.6 | | 15.0 | 15.0 | | 9.6 | 9.6 |
| Project assistants | $0.4 | | 90.9 | 90.9 | | 31.8 | 31.8 |
| Project administrators | $1.1 | | 3.8 | 3.8 | | 4.3 | 4.3 |
| Volunteers | $0.01 | | 930.0 | 930.0 | | 9.3 | 9.3 |
| *Subtotal personnel* | | | | | 101.7 | 120.2 | 120.2 |
| Non-personnel | | | | | | | |
| Expanded beads (kg) | $0.1 | | 1010.0 | | | 111.1 | |
| Curtain fabric (square meter) | $0.2 | | | 998.9 | | | 189.8 |
| Non-personnel, other | $1.0 | | 108.6 | 110.1 | | 108.6 | 110.1 |
| *Subtotal non personnel* | | | | | 0.0 | 219.6 | 299.8 |
| *Total (all resources)* | | | | | **101.7** | **339.8** | **420.0** |

Notation: MDA denotes mass drug administration; EPB denotes expanded polystyrene beads; PIC denotes pyrethroid impregnated curtains; VC denotes vector control; hr denotes hour; kg denotes kilograms, n.a. denotes not applicable (various inputs).

Notes: Total costs equal unit cost times the exact quantity. Volunteers were not paid for vector control activities but occasionally received refreshments. Unit costs for personnel are hourly payments for paid staff, and average hourly refreshment costs for volunteers. Non-personnel other is a composite of all other inputs, including transportation. The quantities shown are rounded.

## LF elimination results and context

Table 6 summarizes the results of the logistic regressions on the effectiveness of the two LF measures. The variable Year2013 is highly significant, indicating marked progress towards elimination in all arms. However, none of the coefficients for interactions between Year2013 and intervention was statistically significant. Furthermore, for each VC arm (VCS and VCI), the coefficients under MfP and AgP were of opposite sign. This lack of consistency and absence of statistical significance indicates that the differences between MDA alone and MDA supplemented by VC initiatives were consistent with chance. That is, at the time and place of the randomized trial, neither VCS nor VCI offered any statistically significant incremental benefit.

As context, Fig 6 and Table G in S1 Text present the prevalence of LF globally, in India, and in the state of Tamil Nadu from 1990 through 2021 based on the GBD study [58]. Over this 31-year period, the prevalence of LF declined by four fifths at all three levels. However, Fig 6A shows that the prevalence rate in India remained about triple the global average. Fig 6B shows that Tamil Nadu's prevalence has been consistently only about a quarter of the rate for India overall. The latest (2021) prevalence rates of 0.7% globally, 2.4% in India and 0.6% in Tamil

**Table 5. Economic cost of interventions (US$) by funder, management level, and type of cost (for 1,000 population).**

| Component | MDA | | EPB | | PIC | |
|---|---|---|---|---|---|---|
| | Gov't | Comm. | Gov't | Comm. | Gov't | Comm. |
| Breakdown by management level | | | | | | |
| 1. State | 208.66 | 0.00 | 0.02 | 0.00 | 0.02 | 0.00 |
| 2. HUD | 3.58 | 0.00 | 1.29 | 0.00 | 1.67 | 0.00 |
| 3. Block | 101.69 | 0.00 | 339.82 | 0.00 | 420.01 | 0.00 |
| 4. Subcenter | 99.24 | 0.00 | 20.34 | 0.00 | 20.34 | 0.00 |
| 5. Village | 0.14 | 66.58 | 0.00 | 129.46 | 0.00 | 361.47 |
| 6. 50 households | 9.30 | 44.36 | 0.00 | 0.00 | 0.00 | 0.00 |
| Subtotal | 422.61 | 110.94 | 361.46 | 129.46 | 442.04 | 361.47 |
| Breakdown by type of cost (government and community combined, with %) | | | | | | |
| Financial costs | 9.30 | (2%) | 451.72 | (92%) | 696.57 | (87%) |
| Non-financial costs | 524.25 | (98%) | 39.2 | (8%) | 106.94 | (13%) |
| Combined total | 533.55 | (100%) | 490.92 | (100%) | 803.51 | (100%) |

Notation: MDA denotes mass drug administration; EPB denotes expanded polystyrene beads; PIC denotes pyrethroid impregnated curtains; Gov't denotes government; Comm. Denotes community; HUD denotes health unit district. Combined total sums costs funded by government and communities.

Nadu translate to 56.9 million, 33.4 million, and 0.49 million people affected globally, in India and Tamil Nadu, respectively. Of all people affected by LF globally, 59% are in India, indicating that LF remains a serious public health problem in India.

By dividing the DALY burden by the prevalence of LF in the GBD study in the last pre-pandemic year, we found that each person-year of LF carried a 0.022 DALY burden. In other words, every 45 persons with LF constitute the loss in value equivalent to one year of good health (i.e., 1/0.022 = 45).

### Incremental cost-effectiveness ratio and summary results

Table 7 derives the ICER of MDA, the one intervention that was significantly effective. MDA significantly improved (lowered) both MfP and AfP compared to initial values 3 years earlier in the study area. The reduction is consistent with the national reductions in LF over a longer period [4]. The ICER for MDA is $112 per DALY averted. This ICER is substantially lower than a reference point of India's $1,350.6 per capita GDP in 2010 US dollars from the World Bank [59].

As a descriptive measure of impact of the conditions, Table E in S1 Text presents the mean by condition for the baseline and final years. Using these prevalence levels, S1 Text presents a sensitivity analysis of cost-effectiveness based on changes in MfP between these two years. Because this analysis creates a sample of just 12 observations (i.e., villages) per condition, it is not able to detect subtle differences between arms. Accordingly, the confidence intervals are wide and none of the differences between conditions were statistically significant.

## Discussion

### Key implications

This study documents substantial progress in controlling LF infection in the study area from 2010 through 2013, as well as over a period of over three decades in Tamil Nadu, across India, and globally. This progress was achieved largely through MDA. Yet, despite MDA, an

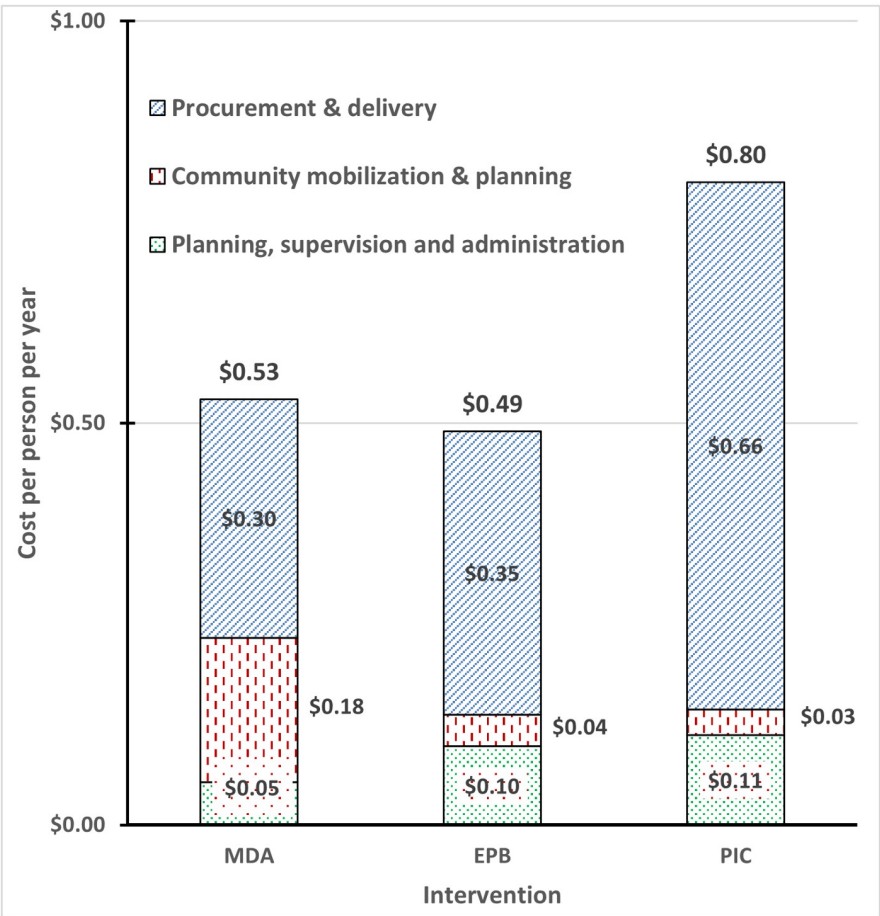

**Fig 2. Cost per person per year by intervention and activity (US$).** Notation: MDA denotes mass drug administration; EPB denotes expanded polystyrene beads); PIC denotes pyrethroid impregnated curtains. The dollar amounts are costs per person per year of the activities. Note that villages receiving EPB also received MDA. Villages receiving PIC also received MDA and EPB.

estimated 33.4 million people in India remained infected by LF in 2021 [58]. A key policy question has been whether existing control strategies are sufficient, or whether VC must be added as well.

Prior to the study period, the impact of MDA was mixed, and transmission assessment survey criteria were not assessed [36]. Tamil Nadu began single-dose (DEC) MDA from 1997–98 in all selected endemic districts and switched to a two-drug regimen (DEC and albendazole) in 2007 [60]. We counted 11 completed MDA rounds in our study villages in Kallakurichi HUD during the 15-year period ending in 2012. However, gaps in annual rounds occurred 1997, 2005 and 2006 in Tamil Nadu, then indicating the potential contributions for complementary strategies, such as VC.

However, our results did not find significant reductions in MfP or AgP within the three-year study period by adding VC to MDA. That is, our current study did not find evidence that VCS or VCI would contribute significantly beyond MDA towards LF elimination as a public health problem. Since the VC interventions added to costs, we concluded that VCS and VCI were not cost-effective additions to MDA during the study period. Continuing with high

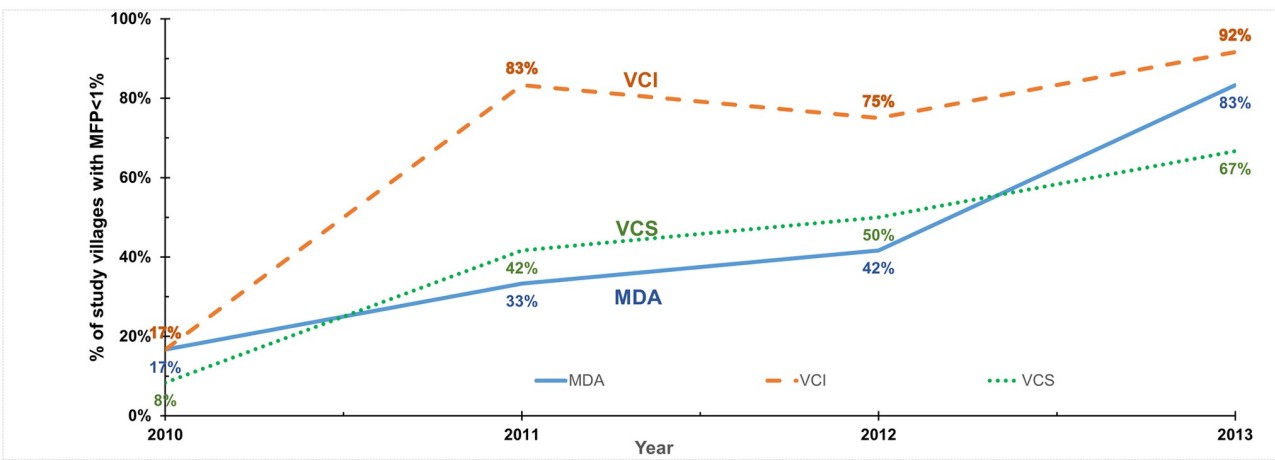

**Fig 3. Share of villages achieving micro-filaria prevalence (MfP) target by intervention arm.** Notes: MfP denotes microfilaria prevalence; MDA denotes mass drug administration; VCS denotes vector control single; VCI denotes vector control integrated; MfP<1% means the village has reached the lymphatic filariasis (LF) elimination threshold.

coverage and full compliance with the regimen of medicines (MDA), rather than adding VC, proved to be the most effective use of limited resources.

## Benefits and evolution of VC

Nevertheless, this study identified some additional benefits from VC. Vector control significantly reduced mosquito biting. From baseline to 2013, VCS and VCI reduced mosquito density by 56% and 77%, respectively, compared to 20% for MDA [28]. Comments during informal community interactions confirmed villagers' awareness of this improvement. Also, curtains can protect households from other insects in addition to mosquitoes, thereby reducing transmission risks for malaria, arboviruses, leishmaniasis and other infections. As VC is

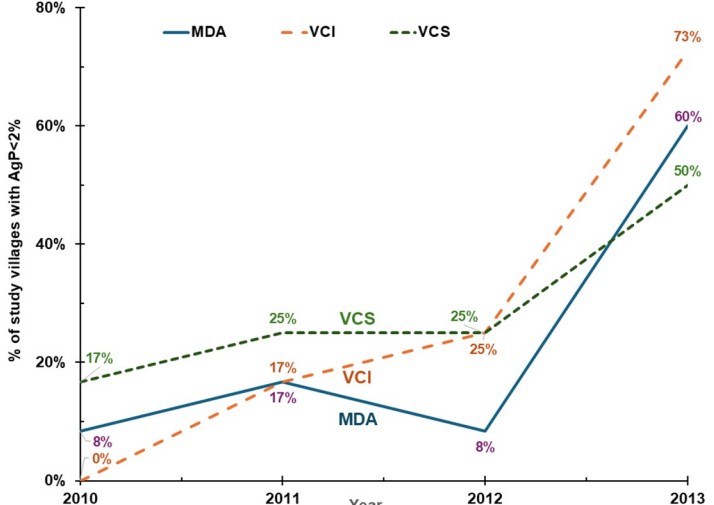

**Fig 4. Share of villages achieving antigen prevalence (AgP) target by intervention arm.** Notes: MDA denotes mass drug administration; VCS denotes vector control single; VCI denotes vector control integrated.

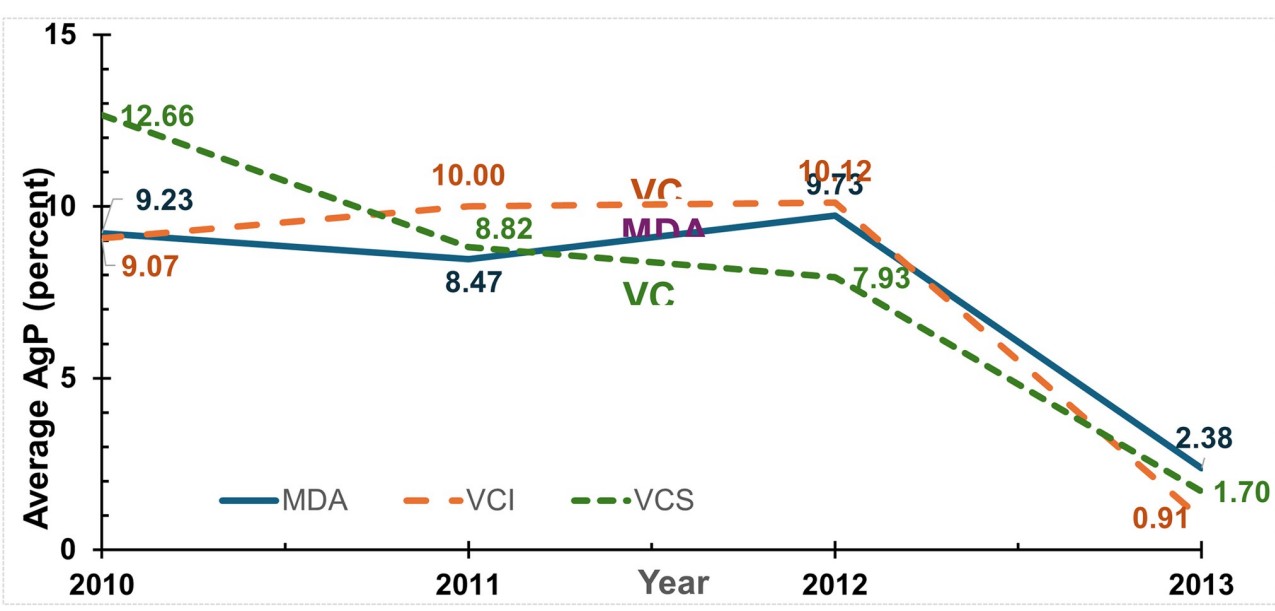

**Fig 5. Average antigen prevalence (AgP) by intervention arm.** Notes: MDA denotes mass drug administration; VCS denotes vector control single; VCI denotes vector control integrated.

**Table 6. Linear regression results of incremental impact of vector control.**

| Independent variable | Coefficient | Standard error | z | Stat. sig. | 95% CI (lower, upper) | |
|---|---|---|---|---|---|---|
| A. Dependent variable: Microfilaria prevalence (MfP) | | | | | | |
| Year2013 | -3.1333 | 0.7020 | -4.46 | 0.00 | -4.5219 | -1.7446 |
| VCS | -0.1223 | 0.7020 | -0.17 | 0.86 | -1.5109 | 1.2664 |
| Year2013 x VCS | 0.3679 | 0.9928 | 0.37 | 0.71 | -1.5960 | 2.3317 |
| VCI | -1.1430 | 0.7020 | -1.63 | 0.11 | -2.5317 | 0.2456 |
| Year2013 x VCI | 0.6418 | 0.9928 | 0.65 | 0.52 | -1.3220 | 2.6057 |
| Constant | 3.8502 | 0.4964 | 7.76 | 0.00 | 2.8682 | 4.8321 |
| B. Dependent variable: Antigen prevalence (AgP) | | | | | | |
| Year2013 | -5.6983 | 2.3784 | -2.40 | 0.02 | -10.4041 | -0.9925 |
| VCS | 4.1813 | 2.2678 | 1.84 | 0.07 | -0.30556 | 8.668 |
| Year2013 x VCS | -4.8591 | 3.3216 | -1.46 | 0.15 | -11.4312 | 1.7128 |
| VCI | 2.8138 | 2.2678 | 1.24 | 0.22 | -1.6731 | 7.3006 |
| Year2013 x VCI | -2.9325 | 3.2863 | -0.89 | 0.37 | -9.4345 | 3.5694 |
| Constant | 8.0779 | 1.6035 | 5.04 | 0.00 | 4.9053 | 11.2506 |

**Notes:** Each panel is based as cross-section time series data with 144 observations (36 villages times 4 years). Mass drug administration (MDA) serves as the reference; VCS denotes vector control singular, using only expanded polystyrene beads; VCI combines VCS with pyrethroid impregnated curtains; CI denotes confidence interval; Comparator for MDA is no intervention; comparator for VCI and VCS is MDA; Independent variable Year2013 shows the impact of MDA in 2013 relative to the reference year (2010). VCS and VCI show the difference in average prevalence levels of villages in these arms compared to those in the MDA arm. The interaction terms show the impacts of vector control compared to MDA alone in 2013. Negative coeffients indicate favorable impact (lower prevalence) toward the eliminiation of LF. Stat. sig. denotes statistical signficance (two-sided) based on normal distribution (z) for the coefficient. R-squared is 0.31 for regression A (MfP) and 0.28 for regression B (AgP).

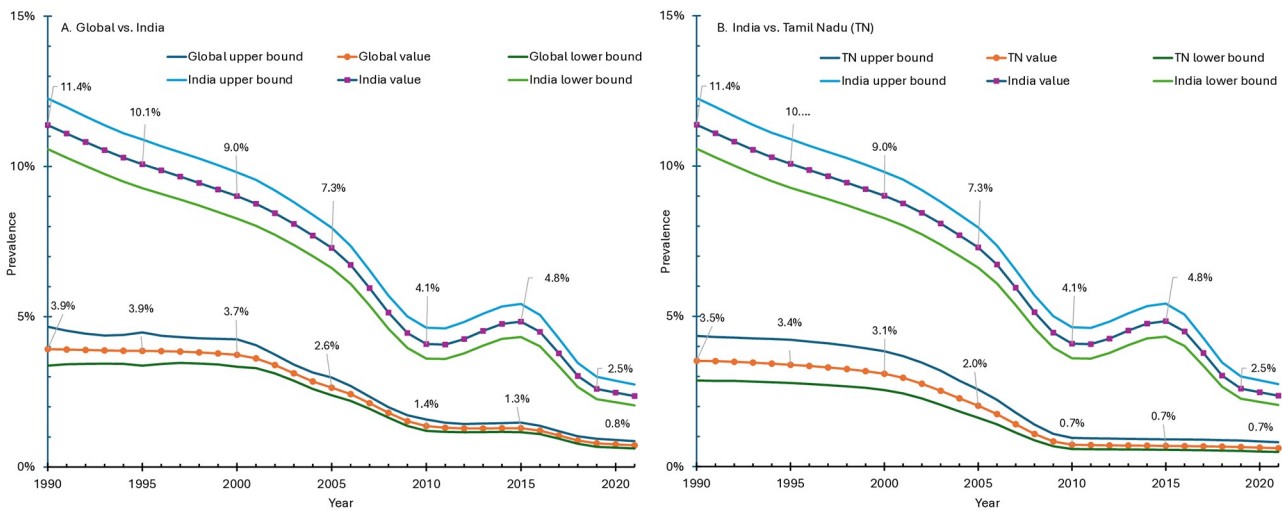

**Fig 6. Prevalence of lymphatic filariasis globally, in India, and in Tamil Nadu, 1990–2021.** Notes: Derived from Global Burden of Diseases study [58]. See Table D in S1 Text for data.

**Table 7. Derivation of the incremental cost-effectiveness ratio of MDA (2010 US dollars).**

| Row | Component | Source | Value |
|---|---|---|---|
| (1) | Baseline (2010) % of villages with MfP<1% | Fig 3 | 17% |
| (2) | Final (2013) % of villages with MfP<1% | Fig 3 | 83% |
| (3) | Cumulative change in % of villages with MfP<1% | (2)–(1) | 66% |
| (4) | Average annual increase in % of villages with MfP<1% | (3) / 3 | 22.0% |
| (5) | Baseline (2010) % of villages with AgP<2% | Fig 4 | 8% |
| (6) | Final (2013) % of villages with AgP<2% | Fig 4 | 60% |
| (7) | Cumulative change in % of villages with AfP<2% | (6)–(5) | 52% |
| (8) | Average annual increase in % of villages with AfP<2% | (7) / 3 | 17.3% |
| (9) | Pooled annual increase in % of villages with MfP<1% and AfP<2% | ((4) + (8)) / 2 | 19.7% |
| (10) | Implicit comparator to MDA | Text | No prevention |
| (11) | Incremental % of villages with MfP<1% and AgP<2% | (9)–(10) | 19.7% |
| (12) | DALYs per person with lymphatic filariasis (LF) | Text | 0.022 |
| (13) | DALYs per 1,000 persons with LF | 1,000 x (12) | 22 |
| (14) | DALYs averted per 1,000 persons in arm | (11) x (13) | 4.33 |
| (15) | Overall cost per 1,000 population | Table 5 | $533.55 |
| (16) | Incremental cost per 1,000 population | (15)–(10) | $533.55 |
| (17) | Incremental cost-effectiveness ratio (ICER), $/DALY | (16) / (14) | $112 |
| (18) | India's 2010 per capita Gross Domestic Product (GDP) in 2010 US$ | [59] | $1,350.6 |
| (19) | ICER as fraction of India's 2010 per capita GDP | (17) / (18) | 8% |

Notes: DALY denotes disability adjusted life years; ICER denotes incremental cost-effectiveness ratio; MDA denotes mass drug administration; MfP denotes microfilaria prevalence in vllage during the year; AgP denotes antigen prevalence in the village during the year.

less dependent on individual behavior than MDA, it could potentially achieve higher coverage and compliance. Both VC interventions were reported to heighten the public's awareness of LF and capacity to control it, and perhaps improved compliance with MDA [28]. However, unless these added benefits were translated into DALYs averted and proved to be substantial, they may not justify the incremental expense of VC.

There could also be alternative meritorious VC interventions. Since the end of data collection in our study, policy makers suggested additional VC scenarios [61]. Evaluating the evidence, a comprehensive review found one published study on the cost-effectiveness of integrating vector control with MDA [62]. That five-year study (from 2002) utilized a combination of larvicides and larvivores for vector control and reported a two-fold increase in cost when compared to standalone MDA interventions, qualitatively similar to our study's results.

If there were a future role for VC to augment MDA, it would likely be strategically targeted transmission hotspots or areas with higher baseline endemicity. Variations in baseline prevalence data (Fig C in S1 Text) show the existence of feasibility of identifying such areas. One VCI data point had a notably high baseline MfP prevalence while another had relatively high endline MfP and AgP prevalence rates. Such variations in coverage also suggest a promising topic for further research applicable to both VC and MDA. These outliers warrant further investigation to understand any unique circumstances that may have contributed to these observations and approaches to avoid or address such hot spots, whether with VC, MDA, or both.

If MDA alone were unable to reduce LF prevalence in some implementation units, further research might be able clarify whether VC would be an effective and cost-effective complement to MDA in such sites. However, for sufficient statistical power, such research would likely require a scale at least comparable to the present study, entailing 12 villages per arm and around 2,000 persons with 3 years of prevalence data (1 year of baseline and 2 years of follow up) per village.

## Evolution of MDA: Increasing adherence

In 2000, long before our study, an evaluation of Tamil Nadu's MDA program found only mediocre adherence, with 30% of households being missed and 46% of those covered not swallowing the recommended MDA tablets [56]. In the decade since data collection for this study ended, MDA has achieved steady progress towards LF elimination. In 2011, WHO refined its protocols for LF chemotherapy [63]. These refinements indicated that after five annual rounds of MDA, an implementation units could suspend MDA if the site had achieved effective (i.e., ≥65%) program coverage during each of those rounds and transmission assessment surveys confirmed that LF prevalence levels at target (MfP<1% or AgP<2%). WHO also recommended post-treatment surveillance to ensure that LF transmission remained interrupted.

Our study data found that MDA was effective and highly cost-effective in controlling LF and progressing towards elimination. Improvements in adherence may help explain the success of MDA in the study site compared to earlier expectations. As MDA is delivered by community-level workers with support from higher levels in the health system, its efficacy reflects success of Tamil Nadu's system of social mobilization and human development, consistent with the State's ranking as one of India's best on this dimension [64]. Reflecting progress, in 2012, Kallakurichi HUD ended its routine MDA program [9, 35]. While we do not have the results of the treatment assessment surveys in the study area or elsewhere in the HUD, the ending of MDA in the HUD, the low levels in the final rounds in study villages with MDA only (Figs 3 and 4), and the low LF prevalence in the state of Tamil Nadu Fig 6B) all indicate that recent MDA rounds achieved effective coverage. Indeed, as of early 2024 the Tamil Nadu was

"on the brink" of eliminating LF and assembling the documentation of official confirmation of this status [65].

Similar evidence of the effectiveness of MDA have extended across India and globally. WHO reported that India as a whole achieved 69.1% MDA coverage in 2023. As this average exceeded the 65% threshold for "effective coverage," repeated rounds of MDA with high coverage should be sufficient to eliminate LF as a public health problem.[9] To increase the effectiveness of MDA, the two-drug regimen has been replaced by triple-drug therapy including ivermectin in some sites [66]. As of the latest Indian government data, the first phase of its 2024 campaign achieved a 95% MDA coverage rate in 96 districts across 11 states and 138 endemic districts have ceased MDA activities [67].

## Study limitations

Two limitations must be acknowledged. The first concerns possibilities of contextual bias because MDA was a routine, government-run program, while VC was an innovative grant-funded intervention. This context may have favored MDA by understating its costs while overstating costs of VC. As data collection on costs of MDA was retrospective, the process may have missed hidden costs of some extended activities, such as more days for MDA distribution and contributed time from other government departments. Costs of VC activities were collected concurrently with the activities themselves, likely avoiding such omissions.

On the other hand, this contextual difference likely had the opposite effect on outcomes—penalizing MDA while favoring VC. MDA operated within routine management structures and the constraints of existing public funding, so managers lacked the resources to optimize MDA. By contrast, VC was managed by researchers with the authority and funding to implement the program according to their best understanding of the relevant science. Cost effectiveness depends on the relationship of incremental costs to incremental outcomes. These preceding possibilities of bias may have increased our estimates of both the incremental cost and incremental effectiveness of the VC arms. As the possible biases affected both the numerator and denominator of cost-effectiveness ratios, they tended to cancel one another. Thus, our finding that VC added to costs without generating additional impact remains relatively free of bias.

Our second limitation concerns variability and what proved to be limited statistical power for assessing outcomes. While each study arm had about 23,000 residents, it had just 12 villages. As VC operates at the village level, our design was a cluster randomized trial. Since key outcomes could only be assessed at the village level, the study had limited power for small differences. As the successful MDA reduced MfP and AgP prevalence rates over successive years, random variations in small numbers of infections in each village became relatively more important, reducing our study's statistical power for assessing incremental differences between arms. Our study design mitigated this problem insofar as possible. We used data from all over India to estimate DALYs averted based on LF prevalence [58]. The careful balancing of the villages among conditions based on baseline prevalence rates reduced the risk of bias. Nevertheless, with outcomes based on just 12 villages per arm, the study had limited power to assess the incremental benefit of adding VCS to MDA or comparing VCI against VCS.

## Conclusions

Before-and-after data in this trial indicate that MDA was highly effective in controlling LF as measured by both MfP and AgP. This success left little scope for further improvement from VC. With substantially reduced levels of LF, the study did not find any significant additional benefits that VCS or VCI added to MDA over the 3-year study period. This study did not assess whether VC could have replaced MDA in achieving the interruption of transmission or

proven more impactful over a longer timeframe. However, our cost data show that VC would probably remain more expensive than MDA. As most of the costs of VC are financial, VC poses greater challenges of financial feasibility than MDA. As India's 2.7% LF prevalence persists at three times the global average [58], the goal of elimination justifies continuing MDA, increasingly with triple-drug therapy [9, 66] in implementation units where LF rates remain above 2% AgP or 1% MfP thresholds.

## Supporting information

**S1 Text.** Fig. A: Discussion of study plans at the Field Unit. Fig. B: Researchers visit a study village to review lymphatic filariasis control. Table A: Microfilaria prevalence (MfP) by village and year. Table B: Antigen prevalence (AgP) by village and year. Table C: Average prevalence of MfP and AgP in base year (2010) and final year (2013) by condition. Text A: Sensitivity analysis. Table D: Incremental costs and change in MfP prevalence. Table E: Incremental percentage reduction in MfP by condition compared to MDA. Table F: Sensitivity analysis of incremental cost-effectiveness ratios of vector control conditions (dollars per person per year per percentage point change in MfP prevalence). Table G: Prevalence of lymphatic filariasis globally. In India, and in Tamil Nadu, 1990–2021. Fig. C: Distributions of base and final years' LF prevalence by condition.
(DOCX)

## Acknowledgments

We thank: Eric A. Ottesen, PJ Hooper, the late Dominique Kyelem and other personnel of the Task Force for Global Health for thoughtful ideas and logistical and financial support; the late S. Elango, S. Sreedharan, and M. Alamelu of the State Public Health Department of Tamil Nadu; M. Geetha, C. Palanichamy, and other health staff of the Kallakurichi HUD; Muthu Kumar, T. Sekar, T.A. Srinivasan, and other staff of the Ariyur Block PHC for providing key program and financial information; health inspectors, village health nurses, other medical officers, and village presidents, the late R. Rajendran, A. Munirathinam, S. Ravi, and V. Ashok Kumar of ICMR-CRME for providing essential program data during our field visits; the volunteer program workers for their critical contributions to the MDA and VC services underlying this study; and Clare L. Hurley of Brandeis University for editorial support.

## Author Contributions

**Conceptualization:** Donald S. Shepard, Sunish I. Pulikkottil, Natarajan Arunachalam, Brij K. Tyagi, Graham B. White.

**Data curation:** Aung K. Lwin, Sunish I. Pulikkottil.

**Formal analysis:** Aung K. Lwin, Sunish I. Pulikkottil.

**Funding acquisition:** Donald S. Shepard, Brij K. Tyagi, Graham B. White.

**Investigation:** Aung K. Lwin, Sunish I. Pulikkottil, Mariapillai Kalimuthu, Natarajan Arunachalam.

**Methodology:** Donald S. Shepard, Aung K. Lwin, Natarajan Arunachalam, Brij K. Tyagi, Graham B. White.

**Project administration:** Donald S. Shepard, Sunish I. Pulikkottil, Brij K. Tyagi, Graham B. White.

**Supervision:** Donald S. Shepard, Sunish I. Pulikkottil, Brij K. Tyagi.

**Validation:** Donald S. Shepard, Aung K. Lwin.

**Visualization:** Donald S. Shepard.

**Writing – original draft:** Donald S. Shepard, Aung K. Lwin.

**Writing – review & editing:** Donald S. Shepard, Aung K. Lwin, Sunish I. Pulikkottil, Mariapillai Kalimuthu, Natarajan Arunachalam, Brij K. Tyagi, Graham B. White.

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
