## [Decision Letter · Decision Letter 0]

14 May 2024

Dear Prof. Shepard,

Thank you very much for submitting your manuscript "Cost-effectiveness of vector control strategies for supplementing mass drug administration for eliminating lymphatic filariasis in India" for consideration at PLOS Neglected Tropical Diseases. As with all papers reviewed by the journal, your manuscript was reviewed by members of the editorial board and by several independent reviewers. In light of the reviews (below this email), we would like to invite the resubmission of a significantly-revised version that takes into account the reviewers' comments. 

We cannot make any decision about publication until we have seen the revised manuscript and your response to the reviewers' comments. Your revised manuscript is also likely to be sent to reviewers for further evaluation.

Sincerely,

Peter U Fischer

Academic Editor

Paul Mireji

Section Editor

Reviewer's Responses to Questions

**Key Review Criteria Required for Acceptance?**

**Methods**

-Are the objectives of the study clearly articulated with a clear testable hypothesis stated?

-Is the study design appropriate to address the stated objectives?

-Is the population clearly described and appropriate for the hypothesis being tested?

-Is the sample size sufficient to ensure adequate power to address the hypothesis being tested?

-Were correct statistical analysis used to support conclusions?

-Are there concerns about ethical or regulatory requirements being met?

Reviewer #1: (No Response)

Reviewer #2: - Are the objectives of the study clearly articulated with a clear testable hypothesis stated?

Yes

- Is the study design appropriate to address the stated objectives? 

Yes

- Is the population clearly described and appropriate for the hypothesis being tested?

the population is clearly described and appropriate to test the said hypotheses

- Is the sample size sufficient to ensure adequate power to address the hypothesis being tested? 

the sample size for prevalence of infection may be provided

- Were correct statistical analysis used to support conclusions?

Yes

- Are there concerns about ethical or regulatory requirements being met?

No

Reviewer #3: The objectives of the study are clearly stated and the study design is appropriate. The study population is described well and appropriate, and the sample size is sufficient for testing the study hypothesis. Appropriate statical analysis of the data was done to arrive at conclusions.

**Results**

-Does the analysis presented match the analysis plan?

-Are the results clearly and completely presented?

-Are the figures (Tables, Images) of sufficient quality for clarity?

Reviewer #1: (No Response)

Reviewer #2: - Does the analysis presented match the analysis plan?

Yes

- Are the results clearly and completely presented?

Yes with suggestions

- Are the figures (Tables, Images) of sufficient quality for clarity?

Yes

Reviewer #3: Does the analysis presented match the analysis plan? Yes

-Are the results clearly and completely presented? Yes

-Are the figures (Tables, Images) of sufficient quality for clarity? Yes

**Conclusions**

-Are the conclusions supported by the data presented?

-Are the limitations of analysis clearly described?

-Do the authors discuss how these data can be helpful to advance our understanding of the topic under study?

-Is public health relevance addressed?

Reviewer #1: (No Response)

Reviewer #2: - Are the conclusions supported by the data presented?

Yes with suggestion

-Are the limitations of analysis clearly described?

Yes

-Do the authors discuss how these data can be helpful to advance our understanding of the topic under study?

Yes, discussed

-Is public health relevance addressed?

Yes

Reviewer #3: Conclusions are supported by the data presented.

Limitations of the study are clearly stated

The authors have explained as to how the data will be helpful to advance the knowledge of the topic of the study.

As stated in the comments, the study has relevance to public health

**Editorial and Data Presentation Modifications?**

Reviewer #1: (No Response)

Reviewer #2: Recommended with minor revision

Reviewer #3: Major revision needed

**Summary and General Comments**

Reviewer #1: The paper presents a cost-effectiveness analysis of vector control strategies for supplementing mass drug administration for eliminating lymphatic filariasis in India.

Overall, the cost data were clearly presented and comprehensive. This data is a valiable contribution to the literature. It would be helpful to report the financial costs as well in the supporting information. 

My major concern is with the regression analysis approach used to capture the effectiveness of the interventions – which needs to be further justified. For example, it is possible that the implementation period was not long enough to show the full long-term incremental benefits of vector control or that vector control should be targeted to transmission hotspots/areas with higher baseline endemicity (both potentially reducing the number of MDA rounds needed in the future). I believe this needs to be made clearer in the discussion. 

In addition, several aspects of the methodology need to be made clearer (such as the perspective and discount rate). I suggest the authors follow the CHEERS checklist.

A sensitivity analysis should be conducted.

Minor comments

In the methods, you say you annualizing capital costs – what discount rate was used

“With MDA offered in all arms, all reduced LF prevalence substantially and significantly from 2010 to 2013”. A little unclear to have both "substantially and significantly"

Value per DALY should be per DALY averted instead - The ICER for MDA is $112 per DALY should be The ICER for MDA is $112 per DALY averted.

Reviewer #2: Detailed comments are attached

Reviewer #3: The manuscript deals with a study on the cost-effectiveness of different interventions for the elimination of lymphatic filariasis which is a major public health problem in many tropical countries. In view of the persisting Mf prevalence even after several rounds, due to sub-optimal coverage and compliance, vector control options including integrated methods are being explored as additional tools. Cost considerations become key issues when adapting multiple interventions and, in this context, the present study is very relevant. The study is well conducted and the cost analysis is done appropriately. I have some major and minor comments which need to be addressed before the manuscript is accepted, which are as below:

Major comments:

1. The conclusion of the study is that the MDA was effective and very cost-effective for

eliminating LF, leaving little scope for vector control for further improvement. This is in contrast to the statement in Authors summary which concludes that the study had limited ability to detect further improvements against the low level of LF endemicity in the study areas. It is expected that when the endemicity is low it will be difficult to discern differences between effectiveness of different control methods. This is a major issue of the study which has a bearing on the conclusion. 

2. Earlier studies have concluded that MDA augmented by VC reduced both the annual transmission potential and the transmission intensity index of LF to zero in Tamil Nadu villages [24], and that when VC was added to MDA decreased the prevalence of microfilaremia by 1% with an added 105 cost of US$0.85 per person annually [25]. But, the present study did not find any additional benefit. This needs to be explained.

3. Lines 499-509: This is another major lacuna. The important data on coverage and compliance of drugs administered, and coverage of breeding habitats for VC which has not been considered. In the absence this data comparing the effectiveness and cost-effectiveness of control tools may not be appropriate.

4. Lines 154-155: The statement should be modified as ‘The World Health Organization defined the eligibility of communities for MDA treatment as the prevalence of filarial parasites (MfP) or antigens of more than 1% and 2%, respectively’.

5. Lines 156-158: What were the base line Mf prevalence and antigen prevalence.

6. Line 181: The drug dosage regimens can be stated.

7. Line 229: Why was the range 350–400 g/m² used?

8. Lines 298-299: Whether the two cost analysis tools: mass drug administration cost analysis tool (MDA-CAT) and VC cost analysis tool (VC-CAT), were validated prior to their use in the study?

9. Lines 356-358: The interventions are targeted against filarial parasite and hence whether is it appropriate to calculate DALYs averted which takes into account the disabilities (lymphedema and hydrocele), that too when the interventions is just for 3 years.

Other minor comments and corrections are indicated in the manuscript uploaded.

PLOS authors have the option to publish the peer review history of their article (what does this mean?). If published, this will include your full peer review and any attached files.

Reviewer #1: No

Reviewer #2: Yes: K.Krishnamoorthy

Reviewer #3: No
---

## [Editor Report · Decision Letter 1]

27 Sep 2024

Dear Prof. Shepard,

Thank you very much for submitting your manuscript "Cost-effectiveness of vector control strategies for supplementing mass drug administration for eliminating lymphatic filariasis in India" for consideration at PLOS Neglected Tropical Diseases. As with all papers reviewed by the journal, your manuscript was reviewed by members of the editorial board and by several independent reviewers. In light of the reviews (below this email), we would like to invite the resubmission of a significantly-revised version that takes into account the reviewers' comments. 

We cannot make any decision about publication until we have seen the revised manuscript and your response to the reviewers' comments. Your revised manuscript is also likely to be sent to reviewers for further evaluation.

Sincerely,

Peter U Fischer

Academic Editor

Paul Mireji

Section Editor
---

## [Editor Report · Decision Letter 2]

28 Oct 2024

PNTD-D-23-01532R2Cost-effectiveness of vector control strategies for supplementing mass drug administration for eliminating lymphatic filariasis in IndiaPLOS Neglected Tropical Diseases Dear Dr. Shepard, Thank you for submitting your manuscript to PLOS Neglected Tropical Diseases. After careful consideration, we feel that it has merit but does not fully meet PLOS Neglected Tropical Diseases's publication criteria as it currently stands. Therefore, we invite you to submit a revised version of the manuscript that addresses the points raised during the review process. Please submit your revised manuscript within 30 days Nov 27 2024 11:59PM. If you will need more time than this to complete your revisions, please reply to this message or contact the journal office at plosntds@plos.org. Please include the following items when submitting your revised manuscript:*
A rebuttal letter that responds to each point raised by the editor and reviewer(s). You should upload this letter as a separate file labeled 'Response to Reviewers'. This file does not need to include responses to any formatting updates and technical items listed in the 'Journal Requirements' section below.*
A marked-up copy of your manuscript that highlights changes made to the original version. You should upload this as a separate file labeled 'Revised Manuscript with Track Changes'.*
An unmarked version of your revised paper without tracked changes. You should upload this as a separate file labeled 'Manuscript'. If you would like to make changes to your financial disclosure, competing interests statement, or data availability statement, please make these updates within the submission form at the time of resubmission. Guidelines for resubmitting your figure files are available below the reviewer comments at the end of this letter. We look forward to receiving your revised manuscript. Kind regards, Peter U FischerAcademic EditorPLOS Neglected Tropical Diseases Paul MirejiSection EditorPLOS Neglected Tropical Diseases

Shaden Kamhawi

co-Editor-in-Chief

Paul Brindley

co-Editor-in-Chief

 **Journal Requirements:** **Additional Editor Comments (if provided):** Minor points stat still need to be addressed:

Line 83, It would make sense to introduce WHO’s Global Programme to eliminate LF (GPELF), before mentioning the national program and to explain the advisory role of WHO for LF elimination. GPELF was first mentioned in line 234.

Line 201, C. quinquefasciatus

Line 205, a brief introduction into the results (what was the question studied in this paragraph) after a long M& M section would be helpful instead of repeating Table 2 shows, Table 3 shows, Table 4 shows, Table 5 shows ….

Line 1984. The abstract (Line 44) explains the study arms MDA, VCS and VCI. But what is PIC? EBP was introduced in line 108, but not PIC. The heavy and inconsistent use of abbreviation makes the ms difficult to read.

Line 314. Why is the explanation of VCS and VCI different from line 44?

Line 520. Are completed MDA rounds also effective MDA rounds with more that 65% coverage?

Lines 524. What were the results of the preTAS and the TAS-1? What were the results of TAS-2 and TAS-3?

Line 527. What is meant by ‘district’? HUD? What was the evaluation unit for TAS (HUD hat 3.5 million residents)?

Line 565. 2023,, VCS….

Line 628. India is already extensively using IDA triple drug MDA.**Reviewers' comments:**   **Figure resubmission:** While revising your submission, please upload your figure files to the Preflight Analysis and Conversion Engine (PACE) digital diagnostic tool, https://pacev2.apexcovantage.com/. PACE helps ensure that figures meet PLOS requirements. To use PACE, you must first register as a user. Registration is free. Then, login and navigate to the UPLOAD tab, where you will find detailed instructions on how to use the tool. If you encounter any issues or have any questions when using PACE, please email PLOS at figures@plos.org. Please note that Supporting Information files do not need this step. If there are other versions of figure files still present in your submission file inventory at resubmission, please replace them with the PACE-processed versions. **Reproducibility:** To enhance the reproducibility of your results, we recommend that authors of applicable studies deposit laboratory protocols in protocols.io, where a protocol can be assigned its own identifier (DOI) such that it can be cited independently in the future. Additionally, PLOS ONE offers an option to publish peer-reviewed clinical study protocols. Read more information on sharing protocols at https://plos.org/protocols?utm_medium=editorial-email&utm_source=authorletters&utm_campaign=protocols

---

## [Editor Report · Decision Letter 3]

20 Nov 2024

Dear Prof. Shepard,

We are pleased to inform you that your manuscript 'Cost-effectiveness of vector control for supplementing mass drug administration for eliminating lymphatic filariasis in India' has been provisionally accepted for publication in PLOS Neglected Tropical Diseases.

Best regards,

Peter U Fischer

Academic Editor

Paul Mireji

Section Editor

Shaden Kamhawi

co-Editor-in-Chief

Paul Brindley

co-Editor-in-Chief

---

## [Editor Report · Acceptance letter]

26 Nov 2024

Dear Prof. Shepard,

We are delighted to inform you that your manuscript, "Cost-effectiveness of vector control for supplementing mass drug administration for eliminating lymphatic filariasis in India," has been formally accepted for publication in PLOS Neglected Tropical Diseases.

Best regards,

Shaden Kamhawi

co-Editor-in-Chief

Paul Brindley

co-Editor-in-Chief
